# Uncertainty Quantification for Fourier Neural Operators

**Tobias Weber, Emilia Magnani, Marvin Pförtner, Philipp Hennig**
Tübingen AI Center, University of Tübingen
`t.weber,emilia.magnani,marvin.pfoertner,philipp.hennig`
@uni-tuebingen.de

## Abstract

In medium-term weather forecasting, deep learning techniques have emerged as a strong alternative to classical numerical solvers for partial differential equations that describe the underlying physical system. While well-established deep learning models such as Fourier Neural Operators are effective at predicting future states of the system, extending these methods to provide ensemble predictions still poses a challenge. However, it is known that ensemble predictions are crucial in real-world applications such as weather, where local dynamics are not necessarily accounted for due to the coarse data resolution. In this paper, we explore different methods for generating ensemble predictions with Fourier Neural Operators trained on a simple one-dimensional PDE dataset: input perturbations and training for multiple outputs via a statistical loss function. Moreover, we propose a Laplace approximation for Fourier layers and show improved uncertainty quantification.

## 1 Introduction

Partial differential equations (PDEs) are used to model complex interactions in real-world phenomena like weather and climate. In recent years, several deep learning methods such as FourCast-Net (Pathak et al., 2022), PanguWeather (Bi et al., 2023), and GraphCast (Lam et al., 2023) have been developed to provide deterministic medium-term PDE predictions. This marks an important step in moving from simulated PDE datasets towards real-world data, which comes with additional challenges such as local weather dynamics that are not resolved in the input data. To address the uncertainties accompanying reanalysis data, ensemble predictions have a long history in weather forecasting. In existing literature, two primary methodologies have been used for adding ensemble predictions to the architectures mentioned above: perturbing the input (Pathak et al., 2022) and training for multiple trajectories via statistical loss function (Lessig et al., 2023). An alternative approach arises from the Bayesian perspective on Deep Learning, by sampling new models from a posterior distribution conditioned on the training data (Gal, 2016). In this context, Laplace approximations have shown to be competitive with more popular alternatives such as variational Bayes or deep ensembles, both in terms of performance and computational cost (Daxberger et al., 2021). While Laplace approximations have been previously explored for Graph Neural Operators (Magnani et al., 2022), its application remains unexplored within the context of Fourier Neural Operators (FNOs). In this paper, we hence focus on FNOs and equip the parameters of the last Fourier layer with an uncertainty estimate. Our primary objective is to compare the resulting ensemble predictions with the existing methodologies mentioned above. Since proper comparison studies of such approaches are missing, we start our analysis in this paper with a simple one-dimensional toy problem of producing ensemble predictions for the trajectories of Korteweg-de Vries equations for different initial states. This stands in contrary to new methods like NeuralGCM (Kochkov et al., 2023) and Gen-Cast (Price et al., 2023) that explore new architectures for explicitly modeling ensemble predictions. Such further comparisons and higher-dimensional experiments are left for future work.

## 2 FOURIER NEURAL OPERATORS

Neural Operators (Kovachki et al., 2023; Li et al., 2021a; 2020b;a; 2021b) are a type of neural network $f_\theta$ - with parameters $\theta \in \mathbb{C}^d$ or $\theta \in \mathbb{R}^d$ - designed to learn solution operators of PDEs. Their training and test sets consist of input-output functions $\mathcal{D} = \{u_n, v_n\}_{n=1}^N$ discretized on some spatio-temporal mesh. The input is usually an initial state $u_n(\cdot, t_0)$, and the output corresponds to the solution of the PDE after a certain time $T$, i.e. $v_n := u_n(\cdot, T)$. Different types of Neural Operator architectures have been developed (Li et al., 2020a;b; 2021a). One prominent method is the so-called Fourier Neural Operator, which consists of pointwise operating linear mappings and so-called Fourier layers. The original Fourier layer (Li et al., 2021a) can be written as

$$\text{FL}(z) = \sigma \left( \mathcal{F}^{-1}(R\mathcal{F}(z)) + Wz \right), \tag{1}$$

where $R$ is a linear transform acting only on the lower Fourier modes and $W$ is a linear mapping operating pointwise in the spatial domain. The higher Fourier modes are truncated, having a similar effect as the non-linear activation function $\sigma$. The restriction to pointwise operations and the transformation into the Fourier domain makes it possible to transfer between different discretizations.

## 3 UNCERTAINTY QUANTIFICATION FOR FNOS

We denote for an input $u_n$ the ensemble of $|K|$ predictions by either $(f_\theta(u_{n,k}))_{k \in K}$, $(f_\theta(u_n)_k)_{k \in K}$ or $(f_{\theta_k}(u_n))_{k \in K}$, highlighting the different positions where the ensemble is introduced. For evaluating an ensemble, we consider its empirical mean $\hat{f}(u_n)$ and the standard deviation $\hat{\sigma}_n$. In the rest of this section, we outline the different approaches and derive the Laplace approximation for the Fourier layer.

**FNO-Perturbed.** A direct approach for extending any model to generate ensemble predictions is by forwarding a batch of perturbed versions of a single input $u_n$. One way is to sample noise $\epsilon_{x,t} \sim \mathcal{N}(0, \tau^2)$ for each input value of the discretized function state $u_n(x, t)$ (Pathak et al., 2022).

**FNO-Ensemble.** Recently Lessig et al. (2023) suggested training directly for $|K|$ predictions, by optimizing $|K|$ parallel prediction heads via an extended loss function. This loss is given by summing over the squared error loss for each prediction, a statistical loss, and the ensemble spread

$$l(v_n, (f_\theta(u_n)_k)_{k \in K}) = \sum_k |v_n - f_\theta(u_n)_k|^2 + |1 - G_{\hat{f}(u_n), \hat{\sigma}_n}(v_n)|^2 + \sqrt{\hat{\sigma}_n}, \tag{2}$$

where $G$ is the Gaussian cumulative density function.

**Laplace approximation for FNOs.** The Bayesian approach for modeling weight-space uncertainty is based on the observation, that the supervised learning objective - extended by a weight regularizer $r(\theta)$ - can be viewed as summing over the negative log-likelihood and a negative log-prior over the weight vector $\theta$ (Bishop, 2006):

$$\theta^* = \arg\min_{\theta \in \mathbb{C}^d} \sum_{n=1}^N \mathcal{L}(v_n, f_\theta(u_n)) + r(\theta) = \arg\min_{\theta \in \mathbb{C}^d} - \sum_{n=1}^N \log p(v_n | f_\theta(u_n)) - \log p(\theta)$$

The weight-space uncertainty under a Laplace approximation is then described by the (unnormalized) log density

$$\log p(\theta | \mathcal{D}) \approx \log p(\theta^* | \mathcal{D}) - \frac{1}{2}(\theta - \theta^*)^T \left[ -\nabla^2_{\theta\theta} \log p(\theta | \mathcal{D}) \right] (\theta - \theta^*)$$

of the Normal distribution $\mathcal{N}\left(\theta^*, -\nabla^2_{\theta\theta} \log p(\theta | \mathcal{D})^{-1}\right)$ (Daxberger et al., 2021). In the deep learning setting, the optimized weights $\theta^*$ are not necessarily a minimizer of the negative log-posterior and therefore the Hessian might not be positive definite, i.e. a valid inverse covariance matrix. A common positive definite approximation is given by the generalized Gauss-Newton matrix $\mathbf{GGN}(\theta) := J_\theta(f_\theta(u_n))^T \Lambda(f_\theta(u_n)) J_\theta(f_\theta(u_n))$, where $J_\theta(f_\theta(u_n)) := \nabla_\theta f_\theta(u_n)$ and $\Lambda(f_\theta(u_n)) = \nabla^2_{f_\theta(u_n)f_\theta(u_n)} \mathcal{L}(v_n, f_\theta(u_n))$ (Botev, 2020).

However, for computational efficiency, the **GGN** is only calculated using a batch of data points and usually only a subset of weights is modeled probabilistically, where the focus is often set on the weights of the last layer. In the case of FNOs, the last linear map is only applied pointwise to the function and therefore does not capture any global structure. As an alternative, we suggest modeling the complex Fourier space parameters $\theta_{FL} := \text{vec}(R)$ of the last Fourier layer probabilistically. To do so, we consider a Kronecker-factored approximation (K-FAC, cf. Martens & Grosse (2015)) to further reduce the cost of computing the **GGN**. This approximation is based on the observation that $J_{\theta_{FL}}(f_\theta(u_n)) := J_{\text{vec}(R\mathcal{F}(z_n))}f_\theta(u_n)J_{\theta_{FL}}R\mathcal{F}(z_n)$ and $J_{\theta_{FL}}R\mathcal{F}(z_n) = \mathcal{F}(z_n)^T \otimes I_{hidden}$. The K-FAC approximation of the covariance matrix can then be derived similarly to Eschenhagen et al. (2023) and Ritter et al. (2018) as follows

$$\Sigma_{\theta_{FL}} \approx \left[ \frac{\sqrt{N}}{B} \sum_{n=1}^{B} \mathcal{F}(z_n)\mathcal{F}(z_n)^T + \tau I \right]^{-1} \otimes \left[ \frac{\sqrt{N}}{B} \sum_{n=1}^{B} b_n \Lambda(f_\theta(u_n))b_n^T + \tau I \right]^{-1}, \quad (3)$$

where we assume a centered Gaussian prior with covariance matrix given by $\tau^2 I$ and denote $b_n := J_{\text{vec}(R\mathcal{F}(z_n))}f_\theta(u_n)$. By sampling new weights $\theta_{FL} \sim \mathcal{N}(\theta_{FL}^*, \Sigma_{FL})$, and passing a single input $u_n$ through the ensemble of resulting models $(f_{\theta_{FL,k} \cup \theta^*})_{k \in K}$, we obtain a corresponding ensemble prediction $(f_{\theta_{FL,k} \cup \theta^*}(u_n))_{k \in K}$. We denote this approach by **FNO-FL**.

## 4 EXPERIMENTAL RESULTS

We consider a one-dimensional FNO with 4 Fourier layers, truncation to 16 modes, and 64 hidden dimensions. It was trained to predict 20 time steps of a solution to the Korteweg-de Vries equation (cf. 4) given the solution at 20 previous time steps.

$$\frac{\partial}{\partial t}u(x,t) = -u(x,t) \cdot \frac{\partial}{\partial x}u(x,t) - \frac{\partial^3}{\partial x^3}u(x,t) \quad (4)$$

The training, validation, and test set consists of 512 solutions for varying initial conditions, each discretized at 256 spatial and 140 time points (Auzina, 2022). For comparing the methods outlined above in the context of model convergence, we consider a short and a long training run. The same training is used for the FNO-Perturbed and FNO-FL, while the FNO-Ensemble is trained for $|K| = 10$ predictions separately. For comparison both short training runs are stopped after the RMSE on the validation set reaches $0.28$, and the long training runs after reaching $0.09$. The $\tau$ parameters in the FNO-Perturbed and FNO-FL are optimized by a simple bisection search and we draw 100 ensemble members.

We consider the following evaluation metrics (Chung et al., 2021):

$$\text{RMSE} = \sqrt{\frac{1}{C} \sum_{n,x,t} (\hat{f}(u_n)(x,t) - v_n(x,t))^2}, \quad (5)$$

$$Q = \frac{1}{C} \sum_{n,x,t} \frac{(f_{\theta^*}(u_n)(x,t) - v_n(x,t))^2}{\hat{\sigma}_{n,x,t}^2}, \text{ and} \quad (6)$$

$$\text{NLL} = \frac{1}{2C} \sum_{n,x,t} \left[ \log 2\pi\hat{\sigma}_{n,x,t}^2 + \frac{(f_{\theta^*}(u_n)(x,t) - v_n(x,t))^2}{\hat{\sigma}_{n,x,t}^2} \right], \quad (7)$$

where $C$ is the number of summands. The RMSE describes the match of the ensemble prediction mean and should be close to zero. The $Q$ estimate indicates the posterior calibration and should be close to one, while the average Gaussian negative-log-likelihood should be low and represents the trade-off between low standard deviations and error terms.

The results of the metrics evaluated on the test set can be found in Table 1. An ensemble prediction for a single test data sample is shown in Figure 1.

## 5 CONCLUSION

While FNO-Ensemble is more expensive to train, FNO-Perturbed and FNO-FL need prior precision tuning. Once a suitable $\tau$ value is found, FNO-FL is an effective tool for quantifying uncertainty

Table 1: Evaluation metrics on 500 training and test samples.

| | Short training run | | | | | | Long training run | | | | | |
| | Training set | | | Test set | | | Training set | | | Test set | | |
| | RMSE | Q | NLL | RMSE | Q | NLL | RMSE | Q | NLL | RMSE | Q | NLL |
|---|---|---|---|---|---|---|---|---|---|---|---|---|
| FNO-Perturbed | 0.26 | 0.83 | 0.26 | 0.25 | 0.83 | 0.25 | **0.08** | 0.85 | -0.59 | 0.08 | **0.99** | -0.52 |
| FNO-Ensemble | **0.25** | 28.71 | 12.39 | 0.25 | 28.62 | 12.34 | **0.08** | 2.82 | **-1.35** | 0.08 | 3.61 | -1.01 |
| FNO-FL | **0.25** | **1.02** | **-0.01** | **0.24** | **1.01** | **-0.02** | **0.08** | 0.88 | -1.32 | **0.07** | 1.09 | **-1.30** |

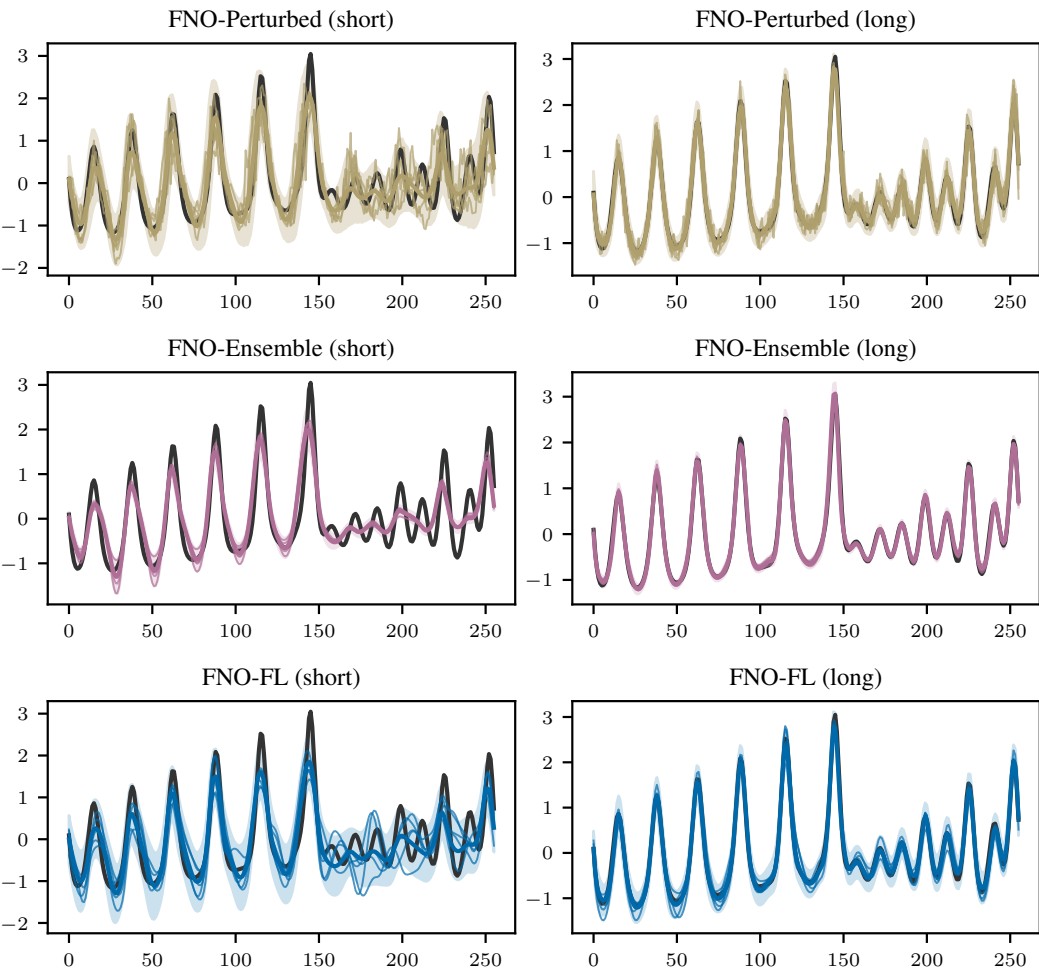

Figure 1: Comparison of FNO-Perturbed (gold), FNO-Ensemble (pink), and FNO-FL (blue) on a single test data sample (black) for both short (left) and long (right) training runs. Each ensemble is depicted by its mean (colored thick line), a 95% c.i. (shaded region) and five members.

in shorter training runs in which the model has not yet converged. However, for longer training runs, its performance is not as clear. In general, the samples from FNO-Ensemble and FNO-FL are considerably more plausible in comparison to the noisy FNO-Perturbed samples.

Note that the uncertainty quantification provided by the Laplace approximation in FNO-FL is arguably strictly more powerful than the other two methods since it is given by a full probability measure on weights as opposed to one approximated by samples.

These promising findings strongly encourage further exploration of ensemble predictions using weight-space uncertainty with Laplace approximations. However, there are still many aspects

to investigate, such as uncertainty in $W$ and scaling these methods to larger networks and high-dimensional data.

ACKNOWLEDGMENTS

The authors gratefully acknowledge financial support by the European Research Council through ERC CoG Action 101123955 ANUBIS ; the DFG Cluster of Excellence "Machine Learning - New Perspectives for Science", EXC 2064/1, project number 390727645; the German Federal Ministry of Education and Research (BMBF) through the Tübingen AI Center (FKZ: 01IS18039A); the DFG SPP 2298 (Project HE 7114/5-1), and the Carl Zeiss Foundation, (project "Certification and Foundations of Safe Machine Learning Systems in Healthcare"), as well as funds from the Ministry of Science, Research and Arts of the State of Baden-Württemberg. The authors thank the International Max Planck Research School for Intelligent Systems (IMPRS-IS) for supporting Tobias Weber, Emilia Magnani, Marvin Pförtner.

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
