# OpenReview forum: "Uncertainty Quantification for Fourier Neural Operators"
_ICLR.cc/2024/Workshop/AI4DiffEqtnsInSci — AI4DiffEqtnsInSci @ ICLR 2024 Poster_

### Official Review · Reviewer_Cdwi · 2024-02-24
**This paper explores different ensemble methods with FNOs for PDE data prediction. This paper is well written. The proposed methods are clearly described and supported by empirical studies.**

**Rating:** 7
**Confidence:** 4

**Review:**

This paper explores three methods for generating ensemble predictions with FNOs: 1) input perturbation, 2) training for multiple outputs via a statistical loss function, and 3) using Laplace approximation for Fourier layers for getting ensemble of weights. Empirical studies are conducted on a simple one-dimensional PDE dataset.

Pros
1. This paper is well organized. The outline of this paper is clear and easy to follow.
2. The empirical studies clearly demonstrate the advantages and the disadvantages of different ensemble methods.
3. The proposed FNO-FL is capable of providing a full probability measure on weights for uncertainty quantification, instead of approximation from generations.

Cons
1. "short run" and "long run" is defined by the validation RMSE. However, since there is no information about the convergence, it is hard to judge if the "short run" is too far from convergence and hence less meaningful. Directly comparing the convergence speed can be a good alternative.
2. The validation RMSE are 0.4 and 0.07 for "short run" and "long run" respectively. However, the training RMSE are around 0.3 and 0.11, and the test EMSE are around 0.27 and 0.1, which show different trends compared with the validation RMSE. The dataset split may cause severe distribution mismatch among training, validation and test sets.

---

### Official Review · Reviewer_UzPe · 2024-02-25
**Good insights on ensemble prediction for Fouier neural operators**

**Rating:** 7
**Confidence:** 4

**Review:**

Summary
------------------

The paper explores ensemble prediction techniques for Fourier Neural Operators (FNOs) in medium-term weather forecasting. It introduces a novel approach using Laplace approximation for uncertainty quantification in FNOs, addressing a gap in existing research. It evaluates various approaches, including input perturbations and statistical loss functions, and introduces a novel uncertainty quantification technique using Laplace approximation for FNOs. Through experiments on a one-dimensional PDE dataset, the paper demonstrates the effectiveness of the proposed method compared to traditional ensemble methods. The findings suggest that the proposed approach improves uncertainty estimation in FNOs, enhancing the reliability of medium-term weather predictions. Further research in ensemble prediction techniques for neural operators is needed, particularly in scaling the methods to larger networks and high-dimensional data. Overall, the paper contributes to advancing the field of ensemble prediction in weather forecasting and related domains.

Evaluation
------------------

**Quality:** The paper is good quality, offering interesting insights on ensembles of FNOs, which could have some impact on weather forecasting using neural operators and other (related) methods.

**Clarity:** The paper provides a clear introduction to the problem domain and previous research efforts. It explains the concepts of neural operators and FNOs concisely and, in my opinion, makes it accessible to general readers, which I appreciate. The methods for UQ and ensemble prediction are well-described, with equations and algorithms nicely structured. Experimental results are presented logically, supported by tables and figures - good visuals overall.

**Originality:** The paper contributes to the field by exploring ensemble prediction techniques for FNOs, an area with limited prior research. So, its originality resides in applying a new method to a challenging problem. The introduction of Laplace approximation for uncertainty quantification in FNOs is somewhat novel, as recognized by the authors and properly cited previous related works, e.g. Magnani et al. The comparison with other ensemble methods adds value to the research, offering insights into the relative performance of different approaches.

**Significance:** The paper's findings have significant implications for weather (and eventually climate?) forecasting and related fields in other domains and disciplines dealing with complex dynamic systems. By improving uncertainty quantification in FNOs, the proposed approach apparently enhances the reliability of medium-term predictions. The evaluation metrics and experimental results demonstrate the practical utility of the method.

Pros
------------------
1. Introduces a novel method using Laplace approximation for UQ in FNOs.
2. Provides a rigorous evaluation of the proposed method through experiments on a one-dimensional PDE dataset.
3. Demonstrates improved uncertainty estimation in FNOs compared to traditional ensemble methods.
4. Enhances the reliability of medium-term weather predictions, addressing a critical need in weather forecasting.
5. Contributes to advancing the field of ensemble prediction techniques for neural operators.
6. Opens avenues for further research in ensemble prediction methods for larger networks and high-dimensional data.

Cons
------------------
1. Requires prior precision tuning for the Laplace approximation, which may add complexity to the implementation. Discussing the limitations would be welcome, yet maybe this could be left for the workshop.
2. Uncertainty quantification in longer training runs using the proposed method appears to be underconfident.
3. Training FNO-Ensemble is computationally expensive compared to other methods. Some comments and ablation study could be added (eventually in the Appendix) and briefly commented (in the main part, just 2 sentences would suffice).
4. The generalization of the proposed method to higher-dimensional datasets needs to be investigated further. Some words of caution about that (substantial) jump to more challenging yet realistic cases would be welcome.
5. The comparison with other SOTA ensemble prediction methods could be more comprehensive. I understand this is not possible in such a short paper format, and consult with the sentence "Such further comparisons and higher-dimensional experiments are left for future work.", some extra lines in Table 1 or appendices wouldn't hurt and would add much value to the analysis.

Clarifications
------------------

- In the intro, "real-world data," I suspect the authors refer to ERA5, which is reanalysis data, not observational data. Please rephrase if so.

- "of input-output functions D = {v_n, u_n}" --> switch to output-input or exchange {u_n,v_n}

- in eq.2 one would expect sigma_n to be constant and not included in the loss, or is it a time-varying process? Please clarify as this would change the solution and ensemble in exponentially varying ways, no?

- clarify in eq.2 to what step prediction K the cdf "G f ˆ (u n ),σ n" refers to.

- is the computational cost an issue for the different approaches? could you include cputime estimates in table 1? i noticed that FNO-Perturbed and FNO-FL perform the same in RMSE but maybe there's other advantage (Q is not a definite score to declare a winner here, esp. in long training runs).

- a discussion on calibration (and eventually some metrics) would be welcome. sentences like "to be underconfident." should be substantiated with numbers - this tool can help: https://github.com/uncertainty-toolbox/uncertainty-toolbox

All these clarifications could be placed in the paper's experimental section or in an appendix/supp.mat section (which I don't find).

Typos & grammar
------------------

- Inconsistent capitalization in "uncertainty quantification" (sometimes capitalized, sometimes not).

- "It exhibits better uncertainty quantification" could be rephrased for clarity (e.g., "it shows improved uncertainty quantification").

- "Moreover, we formulate a new Laplace approximation for Fourier layers" - "formulate" could be replaced with a more precise term like "propose".

- "The training set consists of 520, the validation and test set of 100 solutions" - should include "solutions" after "520" for clarity.

- Inconsistent use of tense throughout the paper (e.g., switching between present and past tense). Choose one and maintain consistency.

- Some sentences are overly long and could be broken down for clarity and readability.

- Punctuation errors, such as missing commas or periods, should be corrected for better readability.

- In the conclusion, "arguably strictly more powerful" could be simplified for clarity.

- clean the bib; missing entries, capital letters. This is a nice tool btw: https://flamingtempura.github.io/bibtex-tidy/.

---

### Meta-Review · Area_Chair_w1sQ · 2024-03-01

**Recommendation:** Accept (Poster)

**Metareview:**

This paper explores the ensemble predictions with Fourier neural operators. Both reviewers agree unanimously. Therefore, I support the acceptance. However, author needs to address the concerns in the camera ready version

---

### Decision · Program_Chairs · 2024-03-01

Accept (Poster)